# Study of distribution and morphological characteristics of the trabecular bone in the uncinate process of the cervical spine using micro-computed tomography

**Yunteng Hao[1‡], Yuan Ma[2‡], Shaojie Zhang[2], Chaoqun Wang[3], Wei Wang[4], Xiaohe Li[2], Shang Gao[2], Kun Li[2], Jie Chen[2], Haiyan Wang[2], Yang Yang[1], Mingjie Gao[2], Jian Wang[2], Zhijun Li[2], Jun Shi[4‡], Xing Wang[2‡]\***

1 Graduate School, Inner Mongolia Medical University, Hohhot, Inner Mongolia, China, 2 Human Anatomy Teaching and Research Section (Digital Medical Center), Inner Mongolia Medical University Basic Medical College, Hohhot, China, 3 Department of Imaging, Affiliated Hospital of Inner Mongolia Medical University, Hohhot, China, 4 Department of Physiology, School of Basic Medicine, Inner Mongolia Medical University, Hohhot, Inner Mongolia, China

‡ YH and YM contributed equally and share first authorship. JS and XW contributed equally and share corresponding authorship.

* wangxing197911@163.com

**Data Availability Statement:** "All relevant data are within the paper and its Supporting Information

## Abstract

The cervical uncinate process is a unique structure of the cervical spine that undergoes significant changes in its morphological characteristics with age, and these changes may be related to osteoporosis. This study aimed to observe the distribution of cancellous bone in the cervical uncinate process and its morphological features using micro-computed tomography (Micro-CT) to gain a deeper understanding of the morphological characteristics of the uncinate microstructure. We performed Micro-CT scans on 31 sets of C3-C7 vertebrae, a total of 155 intact bone samples, and subsequently used the measurement software with the Micro-CT system to obtain parameters related to the cancellous bone of the uncinate process. We found that the cancellous bone of the uncinate process was predominantly longitudinally cross-aligned and continuous with the cancellous bone within the vertebral body. Comparisons between the left and right sides of each parameter showed significant differences only in the bone surface area, and the peaks of each parameter were primarily concentrated in C4-C6. In this study, we found that the C5 uncinate process is the site of most significant stress in the cervical vertebrae, which leads to the earliest occurrence of osteoporosis, and this study provides experimental, theoretical bases for the prevention of cervical spondylosis and osteoporosis, and the diagnosis and treatment of related diseases.

## 1 Introduction

As an essential part of the cervical uncovertebral joint, the cervical uncinate process can lead to compression of peripheral nerves and blood vessels when it suffers from fracture,

files. For additional data, researchers can contact the corresponding author."

**Funding:** The authors received funding from the National Natural Science Foundation of China (grant numbers 81860382); the Program for Young Talents of Science and Technology in Universities of Inner Mongolia Autonomous Region (NJYT22009); the Key Research Projects of Inner Mongolia Medical University (YKD2021ZD011); the Doctoral Scientific Found Project of Inner Mongolia Medical University (YKD2023BSQD014); the Teaching Innovation Team of Inner Mongolia Medical University (NYCXTD202406); the 2024 "Shan Xue" Talent Programme of Inner Mongolia Medical University (ZY20242104). The funder had play an role in study design, data analysis and decision to publish.

**Competing interests:** The authors have declared that no competing interests exist.

hyperplasia, and trauma, which can lead to cervical spine-related lesions such as vertebral artery-type and nerve root-type cervical spondylopathies [1–3]. Current research on the cervical uncinate process focuses on its external morphology [4–6]. However, the cervical uncinate process is crucial for maintaining the stability of the cervical spine, so there is an urgent need for an in-depth study of the microstructural parameters related to the cervical uncinate process.

The cervical uncinate process is the gradual formation as the development of the cervical vertebrae, spinal motion, the overall loading of the cervical spine, and other mechanical requirements increase with age. Bones exhibit macroscopic and microstructural variability in adapting to different mechanical environments in the body, suggesting a correlation between bone structure and stress [7–9]. A comparison of the parameters of the trabeculae in the pedicles, articular processes, vertebral plates, and spinous processes in the lower cervical spine revealed that microstructural differences also existed at different sites [10]. In another study, the articular processes were divided into four regions of interest, and the microstructures were found to differ between sides, vertebral sequences, and regions [11]. Yang et al. [12] divided the vertebral body into dorsal and ventral sides. They found that the volume, volume fraction, and connectivity of dorsal trabeculae were significantly larger than those of the ventral side. In contrast, the separation of dorsal trabeculae was significantly lower than that of the ventral side. Therefore, partitioning and exploring the same bone can allow the microstructural and mechanical changes within different regions of the same site to be observed and reveal its changing characteristics more comprehensively.

The study used Micro-CT to scan the cervical uncinate process to obtain microstructural features and profoundly investigate the correlation between them and the apparent mechanics to assess the risk of degenerative changes and injuries in the cervical uncinate process in the presence of segmental stress overconcentration. It provided a theoretical basis for improving cervical spine health and preventing related diseases.

## 2 Materials and methods

### 2.1 Specimen selection

The Human Anatomy Department of Inner Mongolia Medical University provided 31 adult C3-C7 vertebrae, totaling 155 pieces, all of which are necropsy specimens for daily teaching., The specimens were collected from 10 June 2018 to 30 December 2022 based on the research purpose. All specimens were stored in shaded condition at room temperature, excluding bone defects, deformities, and destruction, gender and age unknown, and no information about the recipient. The study was approved by the Biomedical Ethics Committee of Inner Mongolia Medical University (YKD2018031). The statement confirms that all methods were performed per relevant guidelines and regulations.

### 2.2 Scanning method and parameters

The study arranged the cervical vertebrae in the longitudinal direction according to the sequence of C3-C7 vertebrae. It used a Hiscan XM Micro-CT machine (Suzhou Hesfeld Information Technology Co., Ltd.) and its accompanying analysis and measurement software (Hiscan Analyzer). The fixed scanning parameters were as follows: layer thickness 0.05 mm, layer spacing 0.05 mm, single exposure time 50 ms, tube voltage 60 kV, current 134 uA, image matrix 2000*1600, scanning imaging field of view 10 cm*8 cm, pixel size of 0.05*0.05. Then, the study saved the finished images in the Dicom format on a Lenovo P320 workstation (Suzhou Haisfield Information Technology Co., Ltd.). The study selected the cervical uncinate process as the region of interest (ROI), used the built-in measurement software to extract the

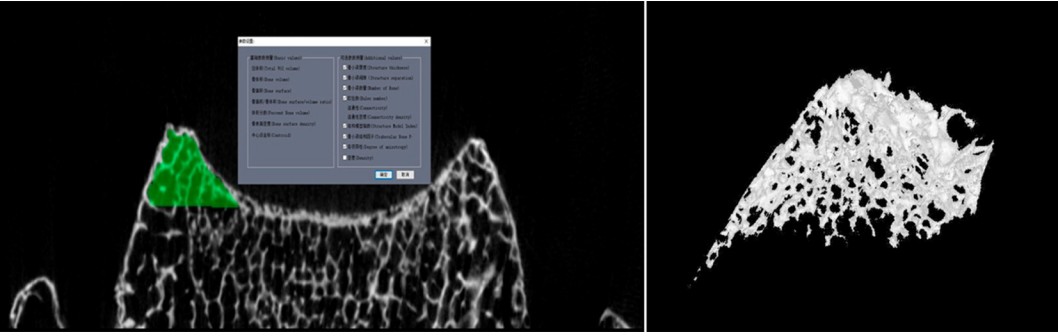

**Fig 1. Selection and three-dimensional reconstruction of trabeculae in the region of interest of the uncinate process.**

microstructure of the ROI, to observe the morphological features of the cervical uncinate process after three-dimensional reconstruction (Figs 1 and 2), and to measure the relevant parameters as shown in Table 1.

## 2.3 Statistical methods

We used the IBM SPSS Statistics for Windows (IBM Corp. Armonk, NY, USA, version 25.0) to perform statistical analysis. The data were expressed as mean ± standard deviation values ($\bar{x} \pm s$). Comparisons between left and right sides were performed by paired-sample t-test; comparisons between different vertebral sequences were performed by one-way analysis of variance (ANOVA), and the LSD method was used to test for homogeneity of variance, while the Kruskal-Wallis test, which is a nonparametric test, was used to test for non-uniformity of variance. The test level $\alpha = 0.05$ was established, and $P < 0.05$ was taken as a significant difference.

## 3 Results

### 3.1 Arrangement and morphological characteristics of the cancellous bone of the uncinate process

After 3D reconstruction, we found that the cancellous bone of the uncinate process was continuous with that of the vertebral body and observed that it was mainly arranged in a longitudinal staggered pattern, which was consistent with the arrangement of the cancellous bone of the vertebral body (Figs 1 and 2).

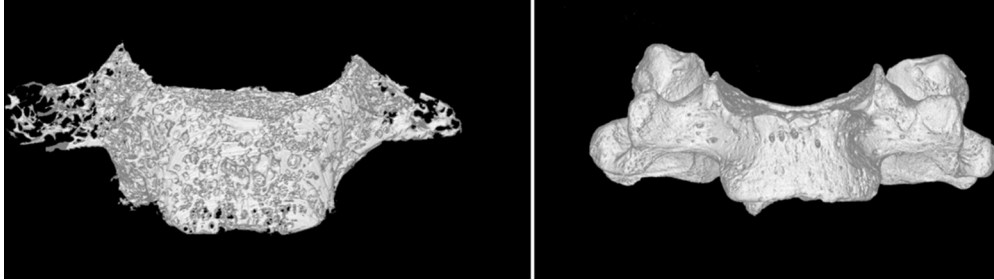

**Fig 2.** Trabecular arrangement of cervical vertebrae under Micro-CT (left), three-dimensional reconstruction results (right).

**Table 1. Morphometric parameters related to the uncinate process of the cervical vertebra.**

| Parameter | Abbreviation | Significance |
|---|---|---|
| Total volume | TV | Volume of the region of interest ($mm^3$) |
| Bone volume | BV | Trabecular bone volume in the region of interest ($mm^3$) |
| Bone surface | BS | Trabecular bone surface area in the region of interest ($mm^2$) |
| Bone surface / bone volume | BS/BV | Indicates the area size of bone tissue per unit volume (1/mm) |
| Bone volume / tissue volume | BV/TV | The ratio of bone volume to tissue volume |
| Bone surface / total volume | BS/TV | Also known as bone surface area density, it can indirectly reflect bone mass (1/mm) |
| Trabecular thickness | Tb.Th | The average thickness of trabecular bone (μm); when osteoporosis occurs, the Tb.Th value decreases |
| Trabecular separation | Tb.Sp | The average width of the medullary cavity between trabecular bones (μm); an increase in Tb.Sp indicates increased bone resorption and which can be a sign of osteoporosis |
| Trabecular number | Tb.N | The number of intersections between bone tissue and non-bone tissue within a given length (1/mm); the Tb.N value decreases when osteoporosis occurs |
| Euler number | Eu.N | A measure of trabecular bone connectivity; the greater the Euler number, the greater the degree of osteoporosis |
| Connectivity | Conn | The degree of interconnection between trabecular bone structures (1/mm); when the connectivity reduced, with the bone loss diseases may occur |
| Degree of anisotropy | DA | The average intercept length in the region of interest of the uncinate process is the ratio of the long diameter to the short diameter of the ROI; in the early stage of osteoporosis, the DA of the load-bearing trabecular bone usually increases, and as the osteoporosis worsens, the DA decreases |
| Structure model index | SMI | The ratio of plate-like and rod-shaped trabecular bones; when osteoporosis occurs, plate-shaped bone trabeculae are reduced, rod-shaped bone trabeculae increase, and the SMI value increases |

### 3.2 Comparison of morphometric parameters of the trabeculae bone in the cervical uncinate process between the left and right sides

The BS of morphometric parameters was more significant on the left side than on the right side in all vertebrae except C3 ($P>0.05$). The Tb. Sp was significantly different only on the right and left sides of C6 ($P<0.05$), and there was no significant difference between the sides of the other vertebrae. The left sides did not differ significantly from the right sides of any vertebrae in other parameters, including TV, BV, BS/BV, BV/TV, BS/TV, and Tb. Th, Tb. N, Eu. N, Conn, DA, and SMI (Fig 3).

### 3.3 Comparison of various morphometric parameters of the trabecular bone in the cervical uncinate process between different vertebral sequences

The TV was spiky, with the peak at C5, and there was no significant difference except between C3 and C4, and between C3 and C5 ($P<0.05$). The BV was generally wavy, but its mean value was trending more gently, with the peak at C5, only the C5 differed significantly from the C3, C4, and C7, and the rest were insignificant ($P>0.05$). The Tb.Th, Tb. Sp and Tb. N was flat, with slight differences between the values, and none differed between groups ($P>0.05$). The Eu. N was wavy, with the peak value located at C4, and the C3 differed significantly from the C6 and C4, and the C5 differed significantly from the C6 ($P<0.05$). The Conn was wavy, with the peak located at C6, and there were significant differences between C3 and C5, between C4 and C5 or C6 ($P<0.05$). The DA showed a sharp peak with increasing vertebral order, and the peak was located at C5, but there was no significant difference between the groups ($P>0.05$). The SMI showed a shallow 'V' shape, with the lowest value located at C4, the only significant difference was between C4 and C7 ($P<0.05$), and the other groups were insignificant($P>0.05$) (Fig 4).

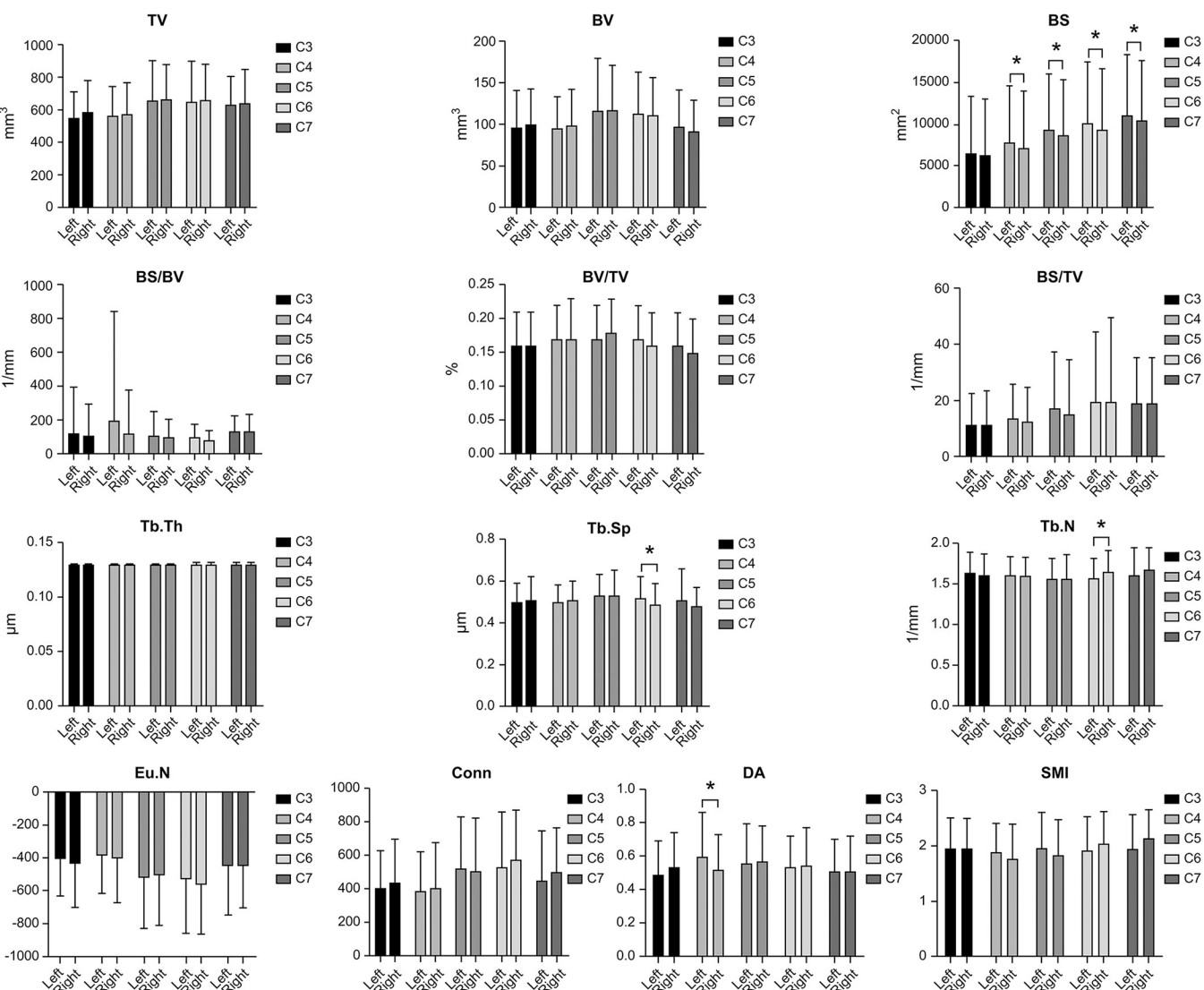

**Fig 3. Comparisons between the left and right sides for the morphometric parameters of the trabecular bone of the uncinate process of the cervical spine.** SMI, structure model index; Tb.Th, trabecular thickness; Tb.Sp, trabecular separation; Tb.N, trabecular number; TV, total volume; BV, bone volume; BS, bone surface; Conn, connectivity; DA, degree of anisotropy; Eu.N, Euler number.

## 4 Discussion

Trabeculae are extensions of the cortical bone within the cancellous bone (i.e., trabeculae are connected to the cortical bone), in mostly spongy or honeycomb shape, aligned in the same direction as the pressure and tension exerted on the bone. They play a role in supporting and reducing weight, cushioning, and containing bone marrow, as well as responding and adapting to deformation [13, 14]. Bone trabeculae are widely distributed in the human body, constituting 80% of the bone surface, and contain various structures such as collagen, minerals, and bone tissue cells, which are the sites of various metabolic processes and an essential origin of the blood system. However, the internal microstructure of bone trabeculae cannot be observed by X-ray, CT, and MR in current imaging examinations. In contrast, the high resolution of Micro-CT can observe the pattern of trabecular arrangement and morphological characteristics of bone trabeculae in the cortical bone and accurately calculate the metrological and

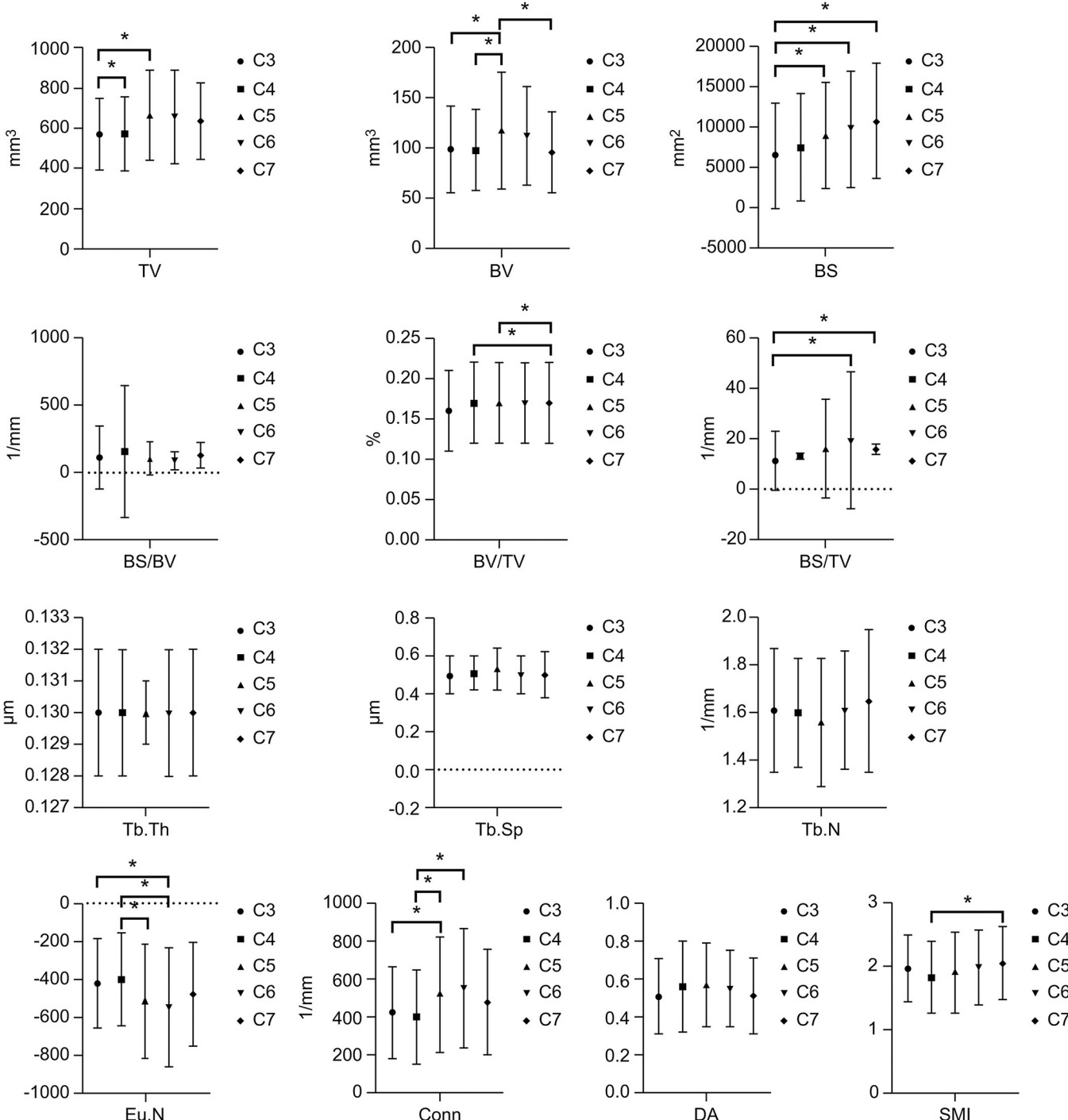

**Fig 4. The trend in the morphometric parameters of the hook vertebra trabecula according to the vertebral number.** SMI, structure model index; Tb.Th, trabecular thickness; Tb.Sp, trabecular separation; Tb.N, trabecular number; TV, total volume; BV, bone volume; BS, bone surface; Conn, connectivity; DA, degree of anisotropy; Eu.N, Euler number.

morphological parameters of the trabeculae in the ROI area by using relevant analysis software [15–18]. Currently, Micro-CT has been used in a large number of applications in the evaluation of bone microarchitecture and has become a routine test in the field of osteoporosis

research through its high-resolution and clear observation of the pattern of bone trabecular arrangement and morphological characteristics within the bone cortex.

Micro-CT also obtains morphological parameters of bone trabeculae by 3D reconstruction and measurement, of which BV/TV, Tb.Th, Tb.N, Tb.Sp, and SMI are currently the most commonly used parameters. The studies have delved into the specific significance of bone microstructural parameters in the characterization of trabecular bone materials, covering both animal and human samples [19–21]. They found that BV/TV is closely related to the apparent maximum stress, force load, elasticity, and stiffness of bone trabeculae, which is the primary indicator for evaluating the trabecular microstructure [22–24]. Yamada et al [25] tested the lumbar spine for compression damage and reported that BV/TV was strongly correlated with vertebral bone strength, but the correlation with Tb.Th was low. In this study, we found that the maximum value of BV/TV of the uncinate process was located at C5, indicating that the stress concentration point of the cervical uncinate process was at C5. Therefore, the C5 uncinate process hyperplasia was earlier and more prevalent, which is consistent with the favourable site of cervical spondylosis due to the uncinate process lesion.

Tb.Th, Tb. Sp and Tb. N constitutes the basic parameters of spatial morphology of bone trabeculae. We perform calculations using the spherical fitting method, where the fitting object is the bone trabeculae in calculating Tb. Th, whereas the Tb. Sp fitting object was the space between the bone trabeculae [26], it can directly reflect the characteristics of bone trabecular changes [27]. In this study, we found that the basic morphological parameters of the bone trabeculae within the uncinate process did not differ significantly between the side aliases and between the vertebrae, suggesting that the morphological characteristics of the bone trabeculae within the uncinate process of each vertebra are stable.

Parameters such as Eu. N, Conn, and DA can indicate the state of bone trabeculae. Peng Jing et al. [28] studied the bone microstructure in the hip fracture region. They found that the interconnected structure of bone trabeculae allowed bone marrow, blood, and other components to reach all parts of the bone smoothly to meet the metabolic growth needs of the organism.

The mechanism of osteoporosis formation also includes a change in the shape of the bone trabeculae, i.e., from plate-like to rod-like [29]. SMI is a parameter that describes the proportion of plate and rod structures in the constituent structures of bone trabeculae. The SMI of bone trabeculae is 0 when plate-like and 3 when rod-like. When osteoporosis occurs, the SMI increases, the rod-like trabeculae increase, the plate-like trabeculae decrease, and the trabeculae become less compression-resistant. Lower SMI tended to occur in the skeleton's weight-bearing region, meaning there are more plate-like bone trabeculae [30]. The minimum value of SMI in this study was at C5, indicating a stress concentration point is in the C5 uncinate process, consistent with previous studies. The maximum value is located at C7, indicating that C7 has a poor loading capacity and is more susceptible to osteoporosis and fractures.

In order to be able to adapt to its mechanical environment, the skeleton of the body may differ in its microstructure between load-bearing and non-load-bearing bone, and blood plays a vital role in bone remodeling processes [31]. In this study, we compared and analyzed the pattern of change of bone trabecular parameters between the right and left aliases of the cervical uncinate process and found that the BS/BV of the left uncinate process was larger, considering that it might be related to the different sources of blood supply between the right and left sides of the uncinate process. The left subclavian artery supplying the left vertebral body and appendages (including the uncinate process) originates from the aortic arch. In contrast, the right subclavian artery supplying the right uncinate process originates from the brachiocephalic trunk, so the left side has a higher blood flow and better blood supply than the right side. In addition, the vertebral artery penetrates upward through the C1-6 transverse foramen.

While the C7 transverse foramen is minor, within which there is only a tiny return of vertebral veins, and the blood supply to nourish the left uncinate process is relatively small, which may be one of the reasons for the most minor BS/TV and Tb. Th of the C7 uncinate process [32–34], and is also objectively determined that C7 is more prone to osteoporosis and degeneration. However, due to the limitations of the scanning coil, Micro-CT can only scan isolated small specimens or live rats under anesthesia and cannot scan dynamic structures in large animals or humans. In addition, this study focused only on the metrological and morphological parameters of the cervical uncinate process trabeculae. It did not compare donor age and sex or different parts of the same vertebrae, which will be the focus of our ongoing exploration.

## 5 Conclusion

In this study, we used Micro-CT to observe the structural characteristics of the bone trabeculae within the uncinate process and summarised the pattern of microstructural changes of the C3-C7 uncinate process. We found that the uncinate process's cancellous bone aligned with the vertebral body's cancellous bone, mainly in a longitudinal staggered arrangement. The C7 uncinate process has poor loading capacity and stress, so the risk of injury in this region is high. The C5 uncinate process is the most stressful area among all the cervical uncinate processes, which leads to the earliest occurrence of hyperplasia in the cervical spine. This study provides a theoretical basis for preventing and treating cervical spondylosis, osteoporosis, and related diseases.

## Supporting information

**S1 Data. The supporting information file is provided in the form of data tables that support the findings reported in the paper.**
(XLS)

## Acknowledgments

The authors thank Suzhou Heathfield Information Technology Co., Ltd. for providing the specimen scanning and data analysis computer used in this research.

## Author Contributions

**Conceptualization:** Shaojie Zhang, Xiaohe Li, Shang Gao, Haiyan Wang, Mingjie Gao, Zhijun Li, Xing Wang.

**Data curation:** Yuan Ma, Chaoqun Wang, Wei Wang, Mingjie Gao, Jun Shi.

**Formal analysis:** Yunteng Hao, Yuan Ma, Jian Wang.

**Funding acquisition:** Xing Wang.

**Investigation:** Yunteng Hao, Yang Yang, Jian Wang.

**Methodology:** Yuan Ma, Shaojie Zhang, Chaoqun Wang, Shang Gao, Kun Li, Jie Chen, Jian Wang.

**Project administration:** Zhijun Li, Jun Shi.

**Resources:** Chaoqun Wang, Wei Wang, Zhijun Li, Xing Wang.

**Software:** Yuan Ma, Xiaohe Li.

**Supervision:** Shaojie Zhang, Haiyan Wang, Mingjie Gao, Zhijun Li, Jun Shi, Xing Wang.

**Validation:** Wei Wang, Shang Gao, Kun Li, Jie Chen, Haiyan Wang, Xing Wang.

**Visualization:** Xiaohe Li, Kun Li, Jie Chen.

**Writing – original draft:** Yunteng Hao, Yang Yang.

**Writing – review & editing:** Yang Yang.

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
