## [Decision Letter · Decision Letter 0]

4 Oct 2024

PONE-D-24-33095Application of Micro-computed Tomography in the Microstructure of Cervical Uncinate Processes Trabecular BonePLOS ONE

Dear Dr. Wang,

Thank you for submitting your manuscript to PLOS ONE. After careful consideration, we feel that it has merit but does not fully meet PLOS ONE’s publication criteria as it currently stands. Therefore, we invite you to submit a revised version of the manuscript that addresses the points raised during the review process.

**ACADEMIC EDITOR:**

**Authors are required to reply all the queries, raised by both the reviewers.**

We look forward to receiving your revised manuscript.

Kind regards,

Priti Chaudhary, M.S.

Academic Editor

PLOS ONE

**Journal Requirements:**

The authors received funding from the National Natural Science Foundation of China (grant numbers 81860382); the Program for Young Talents of Science and Technology  in Universities of Inner Mongolia Autonomous Region(NJYT22009); the Key Research Projects of Inner Mongolia Medical University (YKD2021ZD011); the Doctoral Scientific Found Project of Inner Mongolia Medical University (YKD2023BSQD014); the Teaching Innovation Team of Inner Mongolia Medical University (NYCXTD202406); the 2024 “Shan Xue” Talent Programme of Inner Mongolia Medical University (ZY20242104).

3. In the online submission form, you indicated that All relevant data generated during and/or analyzed during the current study are available from the corresponding author on reasonable request.

5. Please remove your figures from within your manuscript file, leaving only the individual TIFF/EPS image files, uploaded separately. These will be automatically included in the reviewers’ PDF.

Reviewers' comments:

Reviewer's Responses to Questions

**Comments to the Author**

1. Is the manuscript technically sound, and do the data support the conclusions?

Reviewer #1: Yes

Reviewer #2: Yes

2. Has the statistical analysis been performed appropriately and rigorously? 

Reviewer #1: Yes

Reviewer #2: Yes

3. Have the authors made all data underlying the findings in their manuscript fully available?

Reviewer #1: Yes

Reviewer #2: Yes

4. Is the manuscript presented in an intelligible fashion and written in standard English?

Reviewer #1: Yes

Reviewer #2: No

5. Review Comments to the Author

**Reviewer #1:** The is a good work but need some revision as follows;

1. Title

This may ned to be modified as Study of distribution and morphological characteristics of the

trabecular bone in the uncinate process of the cervical spine using micro-computed tomography.

2. Abstract

Background is missing. Just provide one or two sentences here.

3. Introduction

No statement of research problems and justification of the study

Why mechanical structure?

3. Methods

Study area need explain briefly.

How long is the bone stay after the maceration? or Any variations regarding the time of collection or maceration methods.

No repeatability test. You need to determine the methods error.

Inclusion and exclusion criteria were not captured.

Line 102, Excel sheets not table.

Why paired test. The left and right side are two independent structure. So the appropriate test is independent sample test.

Why both ANOVA and Kruskal Wallis in the statistical analyses.

What is the different between text value (α=0.05) and P values?.

Line 134 in summary need to be expunge.

Line 156- scholars?

Limitation and recommendations need to be stated at the end of the discussion.

Trim your conclusion according objectives.

Figure 3, SD in the image, the SD is to high. Please check for possible outlier.

A table need to dedicate to define the landmark used.

Best regards

Thank you

**Reviewer #2: **One glaring issue in the manuscript is poor language. The flow isn't very good and the statements from one section seem to be in another. Some sentences are not clear.

Introduction section isn't structured well, objectives aren't evident.

Results section is not adequate as the descriptive statistics have completely been omitted, and discussion could be structured better too.

Although statistical analysis been performed, the result section doesn't elaborate the output and only states the gist.

The reviewer strongly recommends following STROBE guidelines from EQUATOR website to further align your manuscript. Also, linguistic review seems a must.

Please check the comments in the attached manuscript file.

6. PLOS authors have the option to publish the peer review history of their article (what does this mean?). If published, this will include your full peer review and any attached files.

Reviewer #1: **Yes: **Dr Lawan Hassan Adamu

Reviewer #2: No

---

## [Author Response · Author response to Decision Letter 0]

1 Nov 2024

We are very grateful to your and the reviewers’ critical comments and thoughtful suggestions. Based on these comments and suggestions, we have made careful modification on the original manuscript. All changes made to the text are in red in the revised manuscript so that they may be easily identified. Some of your questions were answered below.

Once again, we acknowledge your comments and constructive suggestions very much, which are valuable in improving the quality of our manuscript. 

Kind regards

Sincerely yours

Xing Wang 

Here are our responses to the reviewers’ comments one-by-one. 

Reviewer #1:

1.Title

This may ned to be modified as Study of distribution and morphological characteristics of the trabecular bone in the uncinate process of the cervical spine using micro-computed tomography.

-Thank you for the title suggested. The precedent version of the title has been replaced.

2.Abstract

Background is missing. Just provide one or two sentences here.

-Thank you for the suggestion. We have added the information suggested by the reviewer.

3.Introduction

3.1No statement of research problems and justification of the study. 

-Thank you for the suggestion. We have added the information suggested by the reviewer.

3.2Why mechanical structure?

-Bone is a dynamic structure that can respond to mechanical stimuli, and this property is known as its mechanical adaptability. The mechanical environment to which bone is subjected is the variety of mechanical forces it is subjected in the body, including pressure, tension, shear, etc., and the effects of these forces on bone structure and function. 

4.Methods

4.1Study area need explain briefly.

-Thank you for the suggestion. We have explained the study area in the manuscript.

4.2How long is the bone stay after the maceration? or Any variations regarding the time of collection or maceration methods.

-The dry bones in the school's anatomical teaching laboratory are usually stored at room temperature in a shady place. According to research by Yang Keqiang and others in the literature, "Study on the Influence Properties of Cortical Bone," storing bones at room temperature does not affect their biomechanical properties.

4.3No repeatability test. You need to determine the methods error.

-In this experiment, we strictly followed the unified equipment standards, performed precise scanning operations on the specimens, and adopted standardized image processing and segmentation algorithms to ensure the reliability of the experimental results. We attached great importance to selecting the region of interest during the experiment to ensure its consistency and accuracy. Also, we provided comprehensive training to the operators to ensure the standardization and consistency of the operation process to minimize methodological errors. In addition, to improve the accuracy of the experimental results further, we performed multiple measurements on the same specimen to reduce the effect of methodological errors.

4.4Inclusion and exclusion criteria were not captured.

-Thank you for the suggestion. We have added the information suggested by the reviewer.

4.5Line 102, Excel sheets not table.

-Thank you for the suggestion. We have added the information suggested by the reviewer.

4.6Why paired test. The left and right side are two independent structure. So the appropriate test is independent sample test.

-A paired-sample t-test for the cervical spine left and right side parameters is appropriate because this test is suitable for comparing differences in means between the same set of samples in two correlated or paired conditions. In this case, the left and right cervical spine parameters for each sample are correlated, so using a paired-sample t-test provides a more accurate assessment of whether there is a statistically significant difference between the two sides.

4.7Why both ANOVA and Kruskal Wallis in the statistical analyses.

-The ANOVA may only be appropriate if the data satisfy a normal distribution or if the variance of the groups is homogeneous. In such cases, the Kruskal-Wallis test is more appropriate as a non-parametric method that does not require these assumptions.

4.8What is the different between text value (α=0.05) and P values?

-The significance level α is a judgment criterion the researcher sets before the statistical test to decide whether to reject the null hypothesis. At the same time, the P-value is a result calculated based on the actual data to measure the probability that the observed data is consistent with the null hypothesis. If the P-value is less than or equal to α, the result is considered statistically significant, and vice versa. 

4.9Line 134 in summary need to be expunge.

-Thank you for your suggestion. We have removed this section. 

4.10Line 156- scholars?

-Thank you for your suggestion. We have revised this section suggested by the reviewer.

4.11Limitation and recommendations need to be stated at the end of the discussion.

-Thank you for your suggestion. We have revised this section suggested by the reviewer.

4.12Trim your conclusion according objectives.

-Thank you for your suggestion. We have revised this section suggested by the reviewer.

4.13Figure 3, SD in the image, the SD is to high. Please check for possible outlier. 

4.14A table need to dedicate to define the landmark used.

-The presence of extreme values in the dataset can increase the standard deviation, and these extremes may be a true reflection of the data rather than an error. We have added notes to the revised version.

Reviewer #2: 

1.One glaring issue in the manuscript is poor language. The flow isn't very good and the statements from one section seem to be in another. Some sentences are not clear.Introduction section isn't structured well, objectives aren't evident. Results section is not adequate as the descriptive statistics have completely been omitted, and discussion could be structured better too. Although statistical analysis been performed, the result section doesn't elaborate the output and only states the gist. The reviewer strongly recommends following STROBE guidelines from EQUATOR website to further align your manuscript. Also, linguistic review seems a must. 

-Thank you for your suggestion. We have tried our best to polish the language in the revised manuscript and have revised the section suggested by the reviewer.

2.Please check the comments in the attached manuscript file.

-Thank you for your suggestion. We have checked the comments in the attached manuscript file.

---

## [Editor Report · Decision Letter 1]

29 Nov 2024

Study of distribution and morphological characteristics of the trabecular bone in the uncinate process of the cervical spine using micro-computed tomography

PONE-D-24-33095R1

Dear Dr. Xing Wang,

We’re pleased to inform you that your manuscript has been judged scientifically suitable for publication and will be formally accepted for publication once it meets all outstanding technical requirements.

Kind regards,

Priti Chaudhary, M.S.

Academic Editor

PLOS ONE
---

## [Editor Report · Acceptance letter]

4 Dec 2024

PONE-D-24-33095R1 

PLOS ONE

Dear Dr. Wang, 

I'm pleased to inform you that your manuscript has been deemed suitable for publication in PLOS ONE. Congratulations! Your manuscript is now being handed over to our production team.

Kind regards, 

on behalf of

Dr. Priti Chaudhary 

Academic Editor

PLOS ONE